# Mapping of c-Fos Expression in Rat Brain Sub/Regions Following Chronic Social Isolation: Effective Treatments of Olanzapine, Clozapine or Fluoxetine

**DOI:** 10.3390/ph17111527

**Published:** 2024-11-13

**Authors:** Andrijana Stanisavljević Ilić, Dragana Filipović

**Affiliations:** Department of Molecular Biology and Endocrinology, “VINČA” Institute of Nuclear Sciences, National Institute of the Republic of Serbia, University of Belgrade, 11000 Belgrade, Serbia; andristanisavljevic@vinca.rs

**Keywords:** c-Fos, rat brain, chronic social isolation, neural circuits, olanzapine, clozapine, fluoxetine

## Abstract

The c-Fos as a marker of cell activation is used to identify brain regions involved in stimuli processing. This review summarizes a pattern of c-Fos immunoreactivity and the overlapping brain sub/regions which may provide hints for the identification of neural circuits that underlie depressive- and anxiety-like behaviors of adult male rats following three and six weeks of chronic social isolation (CSIS), relative to controls, as well as the antipsychotic-like effects of olanzapine (Olz), and clozapine (Clz), and the antidepressant-like effect of fluoxetine (Flx) in CSIS relative to CSIS alone. Additionally, drug-treated controls relative to control rats were also characterized. The overlapping rat brain sub/regions with increased expression of c-Fos immunoreactivity following three or six weeks of CSIS were the retrosplenial granular cortex, c subregion, retrosplenial dysgranular cortex, dorsal dentate gyrus, paraventricular nucleus of the thalamus (posterior part, PVP), lateral/basolateral (LA/BL) complex of the amygdala, caudate putamen, and nucleus accumbens shell. Increased activity of the nucleus accumbens core following exposure of CSIS rats either to Olz, Clz, and Flx treatments was found, whereas these treatments in controls activated the LA/BL complex of the amygdala and PVP. We also outline sub/regions that might represent potential neuroanatomical targets for the aforementioned antipsychotics or antidepressant treatments.

## 1. Introduction

The c-Fos protein is the most widely used functional anatomical marker of cell activity and neuronal circuitry underlying the neuroendocrine, autonomical, and behavioral responses induced by various stimuli, including stress [1,2]. This immediate early gene is rapidly activated by extracellular signals via membrane receptors and ion channels [3]. It encodes a 62 kDa c-Fos protein that dimerizes with the c-Jun protein, forming the AP-1 complex (Activator Protein-1) [4], which functions as a transcription factor promoting the transcription of various genes [5]. The expression of the *c-fos* gene in basal conditions is very low due to the instability of its mRNA and Fos protein-mediated auto-repression of c-Fos transcription [6]. Its transcription occurs rapidly and transiently in the first 5 min, with peak expression approximately 30 min, and peak of c-Fos protein expression approximately 90–120 min following the application of a stimulus [7]. A persistent increase in c-Fos protein levels for at least 24 h has been observed following a variety of stimuli, including chronic social stress [8]. The time for basal expression of *c-fos* mRNA and c-Fos protein also depends on brain regions (some of which express c-Fos at low concentrations or do not express it) [9] and cell activity (cell proliferation, differentiation, or survival). Literature data have shown that different brain regions have different thresholds for c-Fos protein expression, although some brain structures are referred to as constitutive c-Fos-expressing areas [10,11]. Moreover, c-Fos expression has been used to identify brain regions implicated in response to acute or chronic stress [12], and to map neural circuits in response to the stressors [13]. Experimental studies have consistently shown changes in c-Fos expression following social stress [14,15,16]. In addition to neurons, c-Fos expression is noted in glia cells such as astrocytes [17], microglia [18], and oligodendrocytes [19]. Additionally, mapping of c-Fos expression may be used to aid drug classification and predict therapeutic utility [20], as well as to reveal the mechanism of drug action in the brain [21].

Cell activation induced by stress can be altered by drugs. Different classes of drugs, including antipsychotics [22,23] and antidepressants [24,25] may modify c-Fos expression induced by stress. Olanzapine (Olz) and clozapine (Clz) are atypical antipsychotics used to treat depressive and anxiety disorders [26,27], which act as antagonists at dopamine D2 and serotonin 5-HT2A/2C receptors [28]. Both Olz and Clz exhibit a higher affinity for 5-HT2A/2C receptors than for dopamine D2 receptors, a characteristic that contributes to their atypical antipsychotic profiles [29]. As a 5-HT2A/2C antagonist, Olz is believed to provide antidepressant-like effects, alongside its antipsychotic properties. Compared to Clz, Olz has a slightly different receptor binding affinity, whereby its dissociation from the D2 receptor is much slower than that of Clz [30]. Rapid dissociation from D2 receptors and strong antagonism at 5-HT2A/2C receptors are key factors in its effectiveness as an antipsychotic [31]. Since they act on multiple receptors, concurrent antipsychotic and antidepressant treatment is often more effective than antidepressant monotherapy [32,33]. Thus, atypical antipsychotics have become one of the strategies to increase the efficacy of treatment for depression [34], especially treatment-resistant depression. However, the main strategy for treating depression remains antidepressants [35], which are also used to treat anxiety, suggesting that both conditions have some similar neurological processes. Antidepressant fluoxetine (Flx), a selective serotonin reuptake inhibitor, blocks the serotonin 5-hydroxytryptamine transporter (5-HTT), which is responsible for the reuptake of serotonin from the synaptic cleft into presynaptic neurons. This inhibition increases serotonin levels in the synapses, contributing to its antidepressant effects [36]. Also, its active metabolite, norfluoxetine, demonstrates greater selectivity and potency as a 5-HT reuptake inhibitor compared to Flx itself. Notably, norfluoxetine has a significantly longer half-life in comparison to Flx. This extended duration suggests that norfluoxetine may play a crucial role in the therapeutic effects associated with Flx treatment [37]. In addition to blocking 5HTT, Flx also inhibits norepinephrine transporters with a weaker affinity and acts as an antagonist of the serotonin 5-HT2C receptor [38]. It is considered a first-line treatment for depression [39].

Chronic social stress can trigger behaviors in animals that resemble symptoms of depression observed in humans [40]. Psychosocial stress models, such as social defeat stress, are based on the innate social behaviors of individuals. This model involves interactions between two or more subjects, with one establishing a dominant status over the other. After repeated interactions over time, the subordinate subjects display a range of depression-like symptoms, including anhedonia, social withdrawal, reduced locomotor activity, disruptions in the hypothalamic–pituitary–adrenal (HPA) axis and neuropathological changes [41,42,43,44,45,46,47,48]. Importantly, many of these disturbances can be reversed by chronic antidepressant treatments such as Flx and ketamine [49,50,51]. A multifaceted integrated social stress model of depression in female mice has been shown to produce pathological profiles that closely mimic the core symptoms, serum biochemistry, and neural adaptations associated with depression in humans. These profiles can be further modified by systematically administrating a single dose of sub-anesthetic ketamine [48]. Moreover, social isolation has also been utilized to develop the social instability stress model for depression [52,53,54,55], whereby chronic isolation can lead to alterations in gene expression, brain neurochemistry, and behavior [55]. These changes are relevant to the study of human neuropsychiatric disorders. [56,57].

In this review, using mapping of c-Fos expression we provide overlapping brain sub/regions that may provide hints regarding various types of neural circuits in adult male Wistar rats that show depression- and anxiety-like behaviors following three and six weeks of chronic social isolation (CSIS), an animal model of depression. In addition, we compared the three weeks of effective treatments of atypical antipsychotics Olz (7.5 mg/kg/day) or Clz (20 mg/kg/day), as well as the antidepressant Flx (15 mg/kg/day), on the c-Fos expression in brain sub/regions in both controls and CSIS rats. Olz was administered in the last three weeks of six-week CSIS, while Clz and Flx were administered simultaneously with three weeks of CSIS. Our previous studies showed that effective treatments of Clz or Flx prevented CSIS-induced behavior changes in rats [58,59] and increased c-Fos expression, detected by the number of c-Fos immunoreactive cells, in several brain sub/regions compared to CSIS alone [58,60]. Effective treatments of Olz in CSIS rats reversed depressive- and anxiety-like behaviors and decreased the CSIS-induced increase in the number of c-Fos immunoreactive cells in different brain sub/regions [15].

While c-Fos expression serves as a marker of cell activation in response to stimuli during a given time period, it has several limitations. For example, it does not provide information about the frequency or duration of evoked cell activity. As a general activation marker, c-Fos is unable to distinguish between different types of stimuli or specific neurotransmitter pathways, limiting its ability to offer detailed insights into which neural circuits or neurotransmitter systems are involved. Additionally, c-Fos does not differentiate between different cell types, such as excitatory versus inhibitory neurons, nor does it distinguish between neurons and glial cells within the same brain region. Moreover, it does not directly indicate which downstream genes are activated or suppressed in response to chronic stress or antidepressant treatments [61]. Despite these limitations, expression of c-Fos immunoreactivity remains the most commonly used method for determining cell activity (for both neurons and glia) in vivo [62]. As a marker of cell activation, c-Fos has been studied in several experimental animal models since 1990 [61], expanding our understanding of how the nervous system responds to a variety of stimuli, including stressful and pharmacological ones.

## 2. Molecular Mechanisms of c-Fos Expression in Neurons

Stimulation of the cells with extracellular signaling molecules activates c-Fos expression by interactions with many transcriptional regulators, including intracellular calcium (Ca^2+^), mitogen-activated protein kinases (MAPKs), and cyclic adenosine monophosphate response element (CRE) [61,63]. As a component of the transcription factor AP-1, c-Fos regulates late genes directly through the AP-1 binding site where it forms a dimer with c-Jun.

As depicted in Figure 1, the activation of the AP-1 pathway occurs with an influx of calcium into the neuron through the activation of the N-methyl-D-aspartate receptor (NMDAR) or voltage-dependent calcium channel (VDCC) [61,64]. Activation of extracellular signal-regulated kinase (ERK, a member of MAPK cascade) leads to the phosphorylation of regulatory elements such as ETS like-1 protein (Elk1), CRE, serum response factor (SRF), and ribosomal protein kinase S6 (rpS6). These components contribute to the regulation of c-Fos protein synthesis by binding to the serum response element (SRE), a promoter of the *c-fos* gene [65]. cAMP-response element binding protein (CREB) plays a key role in transcription of *c-fos*, itself. For CREB phosphorylation, i.e., activation, several kinases are required, including rpS6, with the most important being mitogen- and stress-activated protein kinase (MSK). Phosphorylated CREB binds to the CRE. Next, the phosphorylated complex of transcription factors binds to the SRE promoter, leading to the transcriptional activation of *c-fos* and its translocation into the cytoplasm of the cell [61].

The synthesized c-Fos protein is then rapidly transported into the cell nucleus after translation, where it forms a heterodimer AP-1 complex with the c-Jun protein (Figure 1), promoting the transcription of various genes. The data reported in the literature also indicate a positive regulation between nuclear-factor kappa-B (NF-κB) activation and c-Fos induction [66]. Thus, in addition to the ERK-Elk-1 and ERK-MSK-CREB signaling pathways that participate in the control of c-Fos expression, the binding of p65 homodimers to the *c-fos* promoter is essential for mouse *c-fos* transcription [66]. Nevertheless, NF-κB activation in the absence of sufficient Elk-1 and/or CREB phosphorylation does not upregulate c-Fos [66]. Moreover, drugs or neurotransmitters that increase calcium elevation or stimulate the cAMP pathway may rapidly activate the *c-fos* gene. Both processes result in the phosphorylation of the transcription factor CREB [67].

## 3. c-Fos as a Marker of Cell Activation

The expression of *c-fos* occurs rapidly and transiently [68] in nearly all cells, as a crucial component of cell signal transduction. Although the expression of c-Fos has been primarily linked with neurons, new research indicates that glial cells may also express this protein. In contrast to neurons, whose *c-fos* transcription is linked to membrane depolarization [69], glial cells (microglia, astrocytes, and oligodendrocytes) produce *c-fos* under the influences of various conditions such as proliferation, differentiation, growth, inflammation, repair, injury, and plasticity [70]. Through the ERK and/or p38 MAPK pathways, inflammatory processes may trigger *c-fos* induction, leading to the phosphorylation of Elk1 at the SRE or CRE sites in the *c-fos* promoter [71]. Also, *c-fos* expression in astrocytes is associated with the stimulation of glutamate receptors such as NMDA [72], in oligodendrocyte progenitors through α-amino-3-hydroxy-5-methyl-4-isoxazolepropionic acid (AMPA) and kainate receptors [73], and in microglia through metabotropic and ionotropic receptors [18]. Glucocorticoids may also directly regulate the *c-fos* gene via their interaction with glucocorticoid receptors in glial cells, facilitating ERK-MAPK signaling pathway [74].

## 4. Chronic Social Isolation as an Animal Model of Depression

It has been shown that social isolation increases the risk of depression by precluding the social stimuli required to modify adaptive responses to new situation [75]. In the CSIS animal model of depression, rats were individually housed in cages with regular auditory and olfactory stimulation but no visual and tactile contacts [76]. CSIS, as a mild psychosocial stress in rats, results in behavioral despair, defined as increased immobility time in the forced swim test, and anhedonia, characterized by decreased ability to experience pleasure from rewarding, reflected as a decrease in sucrose preference, as well as anxiety-like behavior measured by the marble-burying test [77,78]. In addition, CSIS in adult rats can affect cognitive function [79,80]. Rats following CSIS also exhibit neurochemical and neuroendocrine changes [54,81] similar to those observed in humans with psychiatric disorders, including depression [82] and schizophrenia [83], and are alleviated or weakened by pharmacological drugs. Since CSIS displays good face (behavior changes), construct (deregulated HPA axis function, increased proinflammatory responses) [84,85], and pharmacological validity (reversal of depressive symptoms by antidepressants such as Flx), it is considered as a valid animal model of depression [53]. Thus, compromised HPA activity was a consequence of impaired glucocorticoid-mediated feedback inhibition in both hippocampus (HIPP) and prefrontal cortex (PFC) of CSIS rats [84]. Also, increased expression of the prefrontal cortical neural and inducible nitric oxide synthases, along with a concomitant increase in nitric oxide mediated by NF-κB activation and downregulated heat shock protein 70 inducible protein expression, leading to nitrosative stress, was found [86]. Moreover, six weeks of CSIS affected the PFC proteome by downregulating the protein levels involved in the proteasome pathway, glutathione antioxidative system, synaptic vesicle cycle, and endocytosis while upregulating the protein levels of enzymes participating in oxidative phosphorylation [77]. The Flx treatment (15 mg/kg/day) was applied simultaneously with three-week CSIS prevented NF-κB activation and concomitant upregulation of proinflammatory mediators (cyclooxygenase-2, interleukin-1 beta, tumor necrosis factor alpha) in the PFC. Effective Flx treatment in CSIS rats resulted in increased synaptic vesicle dynamic, plasticity, and mitochondrial functionality and a suppression of CSIS-induced impairment of these processes [87]. Moreover, aforementioned rat behaviors are regulated by the activity of neural circuits that control specific physiological functions. CSIS is a phenomenon widespread experienced during the COVID-19 pandemic [88]. Clinically, depressed individuals exhibit a broad range of symptoms, including despair, loss of pleasure, and anxiety [89]. Furthermore, changes in the activity or connection of neural circuitry are associated with depressive behavior [90].

## 5. c-Fos Expression in Rat Brain Sub/Regions Following CSIS

A physiological response to stimuli, including chronic psychosocial stress, results in the activation of various neural circuits that functionally connect distinct brain sub/regions enabling a particular reaction. Thus, three-weeks CSIS led to an increase in the number of c-Fos positive cells in different rat brain sub/regions, such as retrosplenial dysgranular (RSD) and retrosplenial granular cortex, c subregion (RSGc), dorsal dentate gyrus (dDG), paraventricular nucleus of the thalamus posterior part (PVP), lateral/basolateral (LA/BL) complex of the amygdala, caudate putamen (CPu), and nucleus accumbens shell (AcbSh). Six-week CSIS additionally led to increase in the dorsal cornu ammonis (dCA) including dCA1, CA2 and dCA3 subregions and nucleus accumbens core (AcbC) compared to controls [15,58,60]. Increased c-Fos expression was also observed in the subregions of ventral HIPP (vHIPP), medial PFC (mPFC) as well as ventromedial nucleus and dorsomedial nucleus of the hypothalamus following six-weeks CSIS [15]. Moreover, a significant increase in the number of c-Fos positive cells in the frontal cortex of highly aggressive mice susceptible to social stress has been demonstrated [91]. One of the potential reasons could be disrupted redox homeostasis [54] which can result in CSIS-induced rat brain oxidative stress, i.e., excessive reactive oxygen species (ROS) formation caused by mitochondrial dysfunction [92]. Oxidative stress and decrease in capacity of antioxidant defense in the PFC and HIPP of CSIS rats [77,81] are in agreement with the study Shao et al. [93]. Moreover, ROS can trigger the activation of genes and nuclear proteins that function as transcription factors, such as c-Fos [94]. Additionally, a shift in the prooxidant-antioxidant balance caused by CSIS toward a prooxidant state also activates NF-κB, which induces the production of numerous genes involved in the activation of nitrosative and inflammatory mediators such as interleukin-6 (IL-6) [54], potentially triggering c-Fos expression at both mRNA and protein levels in the cells [95,96]. In line with this, increased protein expression of IL-6 has been revealed in chronically socially isolated rats [81,97]. The overlapping rat brain sub/regions with matching pattern of c-Fos expression following three and six weeks of CSIS are presented in Table 1. Identified rat brain sub/regions are implicated in modulating the CSIS response.

The data in the literature revealed different effects of social isolation on cell activation in rodent brains. Following a week of social isolation in adolescent male mice, an increased number of oxytocin/c-Fos positive cells was noted in the paraventricular nucleus of the hypothalamus, which is involved in social behavior [98]. Social isolation stress for four weeks in adult male mice reduced *c-fos* mRNA levels in the HIPP and PFC [99]. In adult male rats, social isolation stress for 30 days increased the number of c-Fos positive cells in the RSC, with no differences in the LA/BL complex of the amygdala compared to the environmental enriched rats [100]. Moreover, 12 weeks of social isolation led to a decrease in the number of c-Fos positive cells in the prelimbic and infralimbic subregions of the mPFC, as well as in the AcbSh, and ventral tegmental area (VTA), with no differences in subregions of dHIPP and AcbC [101].

The schematic connections between the overlapping activated rat brain sub/regions (detected by the increased numbers of c-Fos positive cells) following three and six weeks of CSIS are presented in Figure 2. All these brain sub/regions establish special interconnections (direct or indirect) that are involved in the regulation of social behaviors. The RSGc receives direct input from the subiculum and dCA1 subregion of dHIPP [102], involving both glutamatergic [103] and GABAergic [104], contributing to the memory system. The LA/BL complex of the amygdala sends projections to the RSC [105,106]. The paraventricular nucleus of the thalamus (PVT) has reciprocal connections with both cortical [107] and subcortical brain structures, including the ventral subiculum of HIPP and amygdala [108]. Additionally, it sends dense glutamatergic efferent connections, particularly to the AcbSh [109]. The posterior part of PVT (PVP) has an important role in regulating anxiety-like behaviors [110,111], which may result in increased cell activity following CSIS relative to controls. Thus, the PVT regulates various cognitive and behavioral processes [112] and is included in the neural pathways of stress-related mental disorders [113]. In addition to the mentioned connection, the amygdala and HIPP make the memory circuits [114]. A specific neural circuit from the BLA to the dHIPP has been shown to mediate the effects of repeat stress on hippocampal learning and memory by direct glutamatergic projections [115]. However, the amygdala receives direct hippocampal input only via the vHIPP [116,117,118]. NAc receives glutamatergic afferents from HIPP and amygdala. Particularly, AcbSh plays an integrated role in modulating information processed by the HIPP and amygdala following exposure to emotional stimuli [119]. The CPu coordinates neuronal signals between the cortex, thalamus, and amygdala [120]. Moreover, dorsomedial striatum received projections from the RSC as well [121].

The HIPP is one of the key brain regions involved in stress responses [122] and is associated with mood disorders [123]. Following six weeks of CSIS, the highest percentage increase in c-Fos positive cells was found in the CA2 subregion [15]. This finding may potentially result from the altered function of CA2 in social behavior [124], during CSIS. Also, CSIS-induced c-Fos expression was increased in the dDG after both three and six weeks of CSIS [15,58,60].

The activation of rat amygdala was also revealed following three and six weeks of CSIS. This brain region is responsible for social processing and interaction [125]. A similar pattern of c-Fos expression was noted in the LA/BL complex of the amygdala and subregions of the d/vHIPP which displays strong interconnections with the amygdala. These patterns are consistent with the role of these structures in the stress-induced depression and anxiety [126]. Furthermore, CSIS rats revealed activation of the striatum. The most pronounced activation pattern was observed in the CPu (dorsal striatum) after six weeks of CSIS [15], and significantly less after three weeks of CSIS [58,60] compared to controls. Six weeks of CSIS caused the strongest c-Fos expression in the CPu, as compared to NAc (AcbC and AcbSh), as a component of the ventral striatum. The NAc is involved in the neural circuits that control reward processing and decision-making, whereby dysfunction of this brain region could lead to anhedonia, a core symptom of depression [127]. Moreover, stress that may lead to depressed phenotype, has been shown to alter various functions in NAc, including inhibited dopaminergic activity [102]. Additionally, rats following six weeks of CSIS show an increased number of c-Fos positive cells in NAc [15]. Given that the NAc receives glutamatergic afferents from PFC, BL amygdala and vHIPP [128,129], increased c-Fos expression may be associated with increased cell activity in these brain subregions and their projection to the NAc. Moreover, vHIPP input to the NAc has been shown to regulate behavioral responses to stress in adulthood [44]. In addition, the RSC showed the highest percentage increase in c-Fos positive cells following three weeks of CSIS, with no differences in the percentage of c-Fos positive cells between the RSD and RSGc subregions [58,60].

## 6. c-Fos Expression in Brain Sub/Regions in Control Rats Following Treatments with Olz, Clz, or Flx

Treatments with either Olz, Clz, or Flx in control rats had a significant effect on the c-Fos expression. The overlapping brain sub/regions with matching pattern of c-Fos expression in drugs-treated controls as compared to control rats are shown in Table 2 [15,58,60].

The PVP and LA/BL complex of the amygdala are common sites of action for Olz, Clz, and Flx. In addition to being the site of action for antipsychotic drugs [130], the thalamus contains an abundance of 5HTT [131], which are targets of Flx [132]. Additionally, the PVT is particularly enriched with dopamine D2 receptors [133], for which antipsychotics have a high affinity [134]. Moreover, by interacting with dopamine receptors in various brain regions, Flx may modify the dopaminergic system [135,136]. Hence, the pattern of c-Fos expression may be relevant to the actions of aforementioned drugs.

The amygdala, as a part of the limbic system, that establishes neuronal connections with multiple cortical and limbic brain structures influencing on cognitive processes, affective responses, and social behaviors [137]. Antidepressants, including Flx, modulate amygdala activity, thereby alter functional connectivity and coupling between components of the cortico-limbic system [138,139,140]. The PVP and amygdala, showed the same pattern of c-Fos expression after treatment with different classes of drugs (Figure 3). Coronal brain slices were observed under 4×, 10×, and 20× objectives using a BTC light microscope, equipped with a BIM 313 T digital camera. For each selected brain subregion, we used the Cell Counter Plugin in ImageJ software (version 1.51u) to mark c-Fos positive cells within the defined boundaries of the region of interest. The data in the literature have shown that acute treatment with Clz [141] or Olz in control adult male Sprague Dawley rats preferentially induced c-Fos immunoreactivity in all areas of the central extended amygdala, which is particularly connected to the AcbSh [80]. Moreover, increased number of c-Fos positive cells in AcbSh was also noted in control of adult male Wistar rats treated with chronic treatments (three weeks) of Clz and Olz (Table 2).

Interestingly, Olz treatment in controls showed the highest increase in c-Fos expression in the CPu, while following Clz or Flx treatments, it was observed in RSD. In the striatum, dopamine D1 and D2 receptors subtypes are the most prevalent [142,143], so antipsychotics can specifically target this brain region. Given that Olz has a higher affinity for the D2 receptor compared to Clz [144], whereby both antipsychotics have affinity for 5-HT2A receptor, the balance of affinities of these drugs for D2 and 5-HT2A receptors may influence c-Fos activation in the striatum [145].

RSD, as a part of RSC, is recognized as a brain subregion that integrates information between the thalamus, hippocampal, and neocortical subregions [146]. This region expresses D1-like receptors (D1and D5) [147]. Since activation of D1-like receptors increases the activation of the *c-fos* promoter [148], the increased number of c-Fos positive cells in the RSD of Clz-treated control rats may be related to its action on the D1 receptor [149]. Additionally, since the RSC receives long-range excitatory and inhibitory inputs from dHIPP [150], primarily from the subiculum and secondarily from CA1 [102], the increased c-Fos expression in Flx-treated control rats may be associated with increased dHIPP cell activity and hippocampal projection to the RSC. In all, Flx did not change the behavior phenotype in control rats, so increased expression of c-Fos might indicate adaptive cell responses to chronic Flx treatment.

## 7. c-Fos Expression in Brain Sub/Regions in CSIS Rats After Effective Treatments with Olz, Clz, or Flx

Following effective treatments of Olz, Clz, or Flx in CSIS rats, the activation of the AcbC, a part of NAc, was revealed (Table 3) [15,58,60]. In general, striatal subregions (NAc and CPu) are rich in dopaminergic receptors, a primary target of antipsychotic drugs [151]. Moreover, dopamine acting on D1 and D2 receptors modulates the activity of direct and indirect pathways that regulate the excitability of striatal neurons [152]. Thus, altered dopaminergic innervation of NAc following antipsychotic treatments is involved in selection and integration of excitatory glutamatergic inputs from limbic (basolateral amygdala and vHIPP) and cortical (medial prefrontal cortex) structures which govern behavioral output [153]. In fact, D2 dopamine receptor and 5-HT2A serotonin receptor have been suggested as the primary targets of antipsychotics, including Clz and Olz [154]. Additionally, Flx enhances dopamine function in the NAc through increased D2 mRNA and D2 receptor expression [155]. Furthermore, serotonin acts through several serotonin receptors in the brain regulating dopamine cellular activity and release [156], so the clinical efficacy of Flx which acts on serotonin systems may be due in part to its effects on the dopamine system. It has been noted that majority of serotonin effects on dopamine neurons may be indirect, mediated by actions on complex neural circuitry rather than direct effects on dopamine terminals [156]. Thus, serotonergic projections from dorsal raphe nucleus to the striatum increases dopamine release [157,158]. Additionally, since the BLA provides glutamate input to the AcbC [159,160], increased c-Fos expression in Flx-treated CSIS rats could also be related to increased cell activity in the BLA under the same condition (Figure 4). Also, it was noted that AcbC receives fewer inputs from hippocampal dCA1, compared to AcbSh [161]. Additionally, the PVT send glutamatergic projections to the NAc, with a notable emphasis on the AcbSh [161,162]. In fact, the different patterns of c-Fos expression shown here may be driven by complex interactions within neural circuits, arising from different drug effects on dopamine, glutamatergic and serotonin receptors in different cell types. Moreover, CPu subregion is involved in the regulation of movement [163]. Thus, the increased c-Fos expression in this region could be related to the tendency of antipsychotics to produce extrapyramidal side effects [80,164,165].

All the CA subregions of the dHIPP showed an increased number of c-Fos positive cells in the Flx-treated CSIS rats as compared to CSIS alone (Figure 4 and Figure 5), with the highest percentage expression in dCA3 [58]. In the rat HIPP, dopaminergic D1 receptor activation is also associated with c-Fos expression, and may contribute to hippocampal synaptic plasticity [166]. Moreover, the D1 receptor in the dHIPP has an important role in mediating the antidepressant action of drugs like Flx [167]. It has been shown that activation of D1 receptors by chronic Flx treatment is functionally coupled to adenylyl cyclase, leading to upregulation of cAMP/PKA (protein kinase A) signaling and thus increased neuronal excitability [168]. In contrast to Flx, Olz decreased c-Fos expression in the dHIPP subregions of CSIS rats (Figure 4 and Figure 5), with the highest decline in dCA3, which is richer in the internal connection than other hippocampal subregions [169]. The differences in c-Fos expression in dHIPP between different drugs may be due to variations in experimental design. Furthermore, an additional enhancement of c-Fos expression in Clz- or Flx-treated CSIS rats relative to CSIS alone suggests cumulative effects of the drugs and CSIS in the dCA1 (Table 3) (Figure 4 and Figure 5). Actually, it is possible that the duration of stress (three or six weeks of CSIS) influences dCA1 activity in terms of how various drug classes impact cell activity.

The c-Fos positive cells were more abundant in the PVP following effective Flx and Clz treatments in CSIS rats compared to the CSIS group (Table 3, Figure 4 and Figure 5). Since the PVT is highly sensitive to stress and has the highest density of D2-like receptors, which are the principal target of drug action, this result most likely occurred from the cumulative effect of the drugs and CSIS. In contrast to their effect in the RSC of control rats, Flx and Clz had no effects on c-Fos expression in the CSIS rats (Figure 4). Moreover, RSC showed a decreased number of c-Fos positive cells, following effective Olz treatment in CSIS rats (Figure 4), with no effects in the controls. The interconnections between brain sub/regions along with overlapping patterns of c-Fos expression, following effective Olz, Clz, or Flx treatments in CSIS rats, are presented in Figure 5.

In addition to the proposed mechanisms of action, such s as receptor activation and the downstream effects on c-Fos expression, consideration of other signaling pathways or receptor interactions could provide a more comprehensive view. Alternative mechanisms could involve compensatory responses from other neurotransmitter systems, such as GABAergic or glutamatergic signaling, or the participation of other molecular factors, including ion channels or kinase enzymes, which may also impact neuroplasticity and behavior. Thus, Flx triggers the phosphorylation of ERK1/2 through the activation of 5-HT2B receptors, leading to an increase in c-Fos expression, which subsequently stimulates the production of a brain-derived neurotrophic factor in astrocytes, a key protein essential for neuroplasticity and the formation of new neural connections. [170]. Moreover, Clz’s therapeutic effects and c-Fos expression may involve the regulation of HPA-corticolimbic circuits that are activated during the stress response [171]. Regarding Olz, it binds to a range of receptors, such as histamine, muscarinic, and adrenergic receptors, which contribute to its complex pharmacodynamic effects. This wide receptor activity produces diverse c-Fos expression patterns across different brain regions [172].

Furthermore, there are several studies that investigated the effects of antipsychotics Clz or Olz, or antidepressant Flx on c-Fos expression in the rodent brain. Thus, pretreatment with Clz (5 mg/kg) decreased the number of c-Fos immunoreactive cells in the brain regions that regulate the activity of the HPA axis and prevented social interaction deficits in male rats exposed to acute restraint stress [79]. Pretreatment with Clz (5 mg/kg) in an MK-801-induced mouse model of schizophrenia decreased the number of c-Fos positive cells in the posterior cingulate cortex and RSC [173]. A seven-day Clz treatment (7 mg/kg) in male rats exposed either to an acute stressor (forced swimming episode, FSW) or a 13-day mild unpredictable stressor (with exposure to a novelty stressor-FSW on day 14) revealed differential effects on c-Fos immunoreactivity in brain subregions, indicating that the effect of Clz on cell activity is affected by the duration of stress [174]. Two weeks of Flx (14 mg/kg) treatment in mice ameliorated depressive and anxiety-like behavior along with a significant increase in the number of c-Fos positive cells in DG, a brain subregion involved in neurogenesis [175]. Moreover, seven-day treatment of Flx (10 mg/kg) combined with Olz (5 mg/kg) in rats inhibited the induction of two immediate early gene transcription factors (pCREB and FOS), which are linked to long-term alterations in synaptic efficacy and structure in the piriform cortex, PFC and HIPP. This combination could help in treatment-resistant depression [176].

However, it is worth noting that even though antipsychotic drugs induce changes in certain brain sub/regions, it does not necessarily mean that these sub/regions are targets for controlling neuropsychiatric diseases. Aside from that, most of the extrapyramidal side effects of antipsychotic drugs are caused by strong dopaminergic blockade in the striatum (nigro-striatal pathway) [177]. The antipsychotic effects of atypical antipsychotics (Olz and Clz), which alleviate both positive and negative symptoms, are primarily related to their activities on the mesolimbic (including NAc) and mesocortical pathways (dopaminergic neurons that project from the VTA to cortical regions).

## 8. Conclusions

This review highlights the significance of specific brain regions as components of neural circuits that undergo changes under chronic social isolation stress (CSIS) and respond differently to various antidepressant and antipsychotic treatments. Increased c-Fos immunoreactivity in several overlapping brain regions within the limbic system (including the retrosplenial cortex, hippocampus, amygdala, and thalamus) and basal ganglia (striatum) suggests that these areas are part of the neural networks involved in responses to three and six weeks of CSIS, as well as the development of depressive and anxiety-like behaviors. Additionally, activation of AcbC following exposure of CSIS rats to Olz, Clz, and Flx treatments may be related to their therapeutic effect. In controls, the same treatments activated regions, such as the amygdala and thalamus, indicating potentially different mechanisms of action for these drugs in controls and CSIS. The NAc, with heightened c-Fos activation, could serve as an important neuroanatomical target for the further development and application of antidepressant and antipsychotic treatments, particularly in the context of depression induced by CSIS.

## Figures and Tables

**Figure 1 pharmaceuticals-17-01527-f001:**
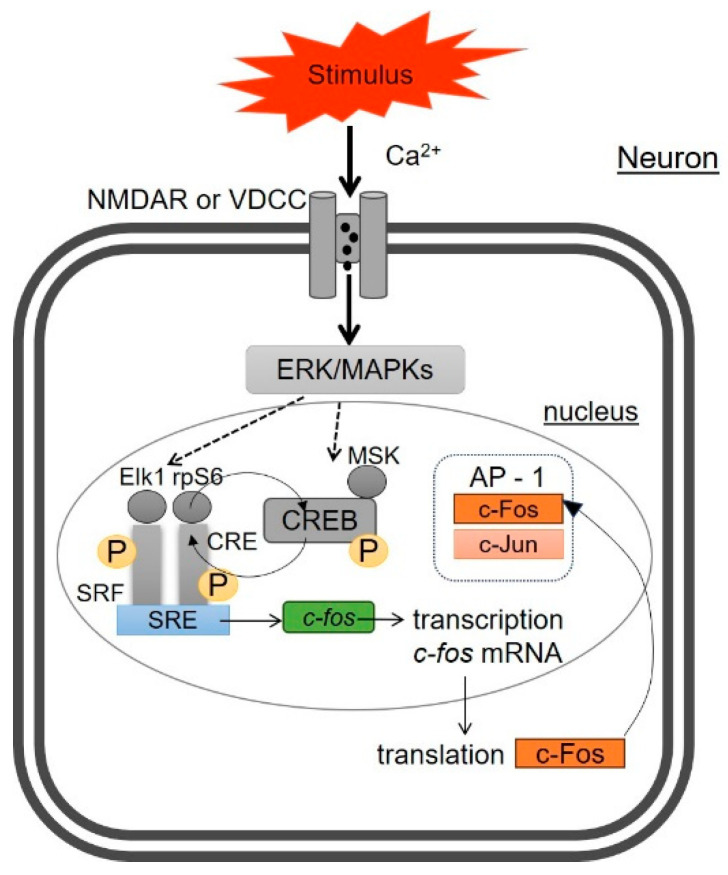
Molecular mechanisms of c-Fos expression.

**Figure 2 pharmaceuticals-17-01527-f002:**
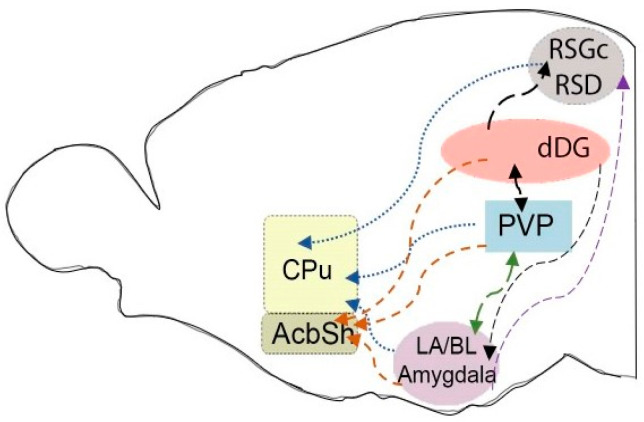
Schematic illustrating neural circuits of the rat brain sub/regions with an increased number of c-Fos positive cells following three and six weeks of CSIS. Brain sub/regions: retrosplenial granular cortex, c region (RSGc), retrosplenial dysgranular cortex (RSD), dorsal dentate gyrus (dDG), paraventricular nucleus of thalamus, posterior part (PVP), lateral/basolateral (LA/BL) complex of the amygdala, caudate putamen (CPu), and nucleus accumbens shell (AcbSh).

**Figure 3 pharmaceuticals-17-01527-f003:**
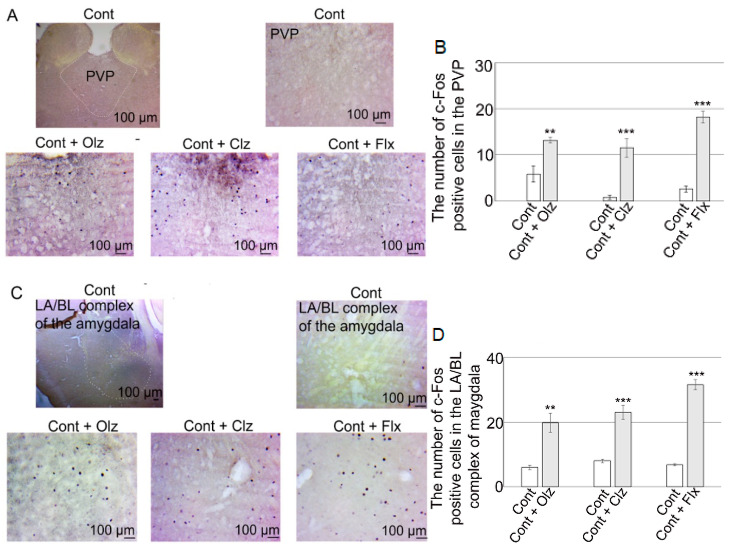
The representative photomicrographs of coronal brain slice with c-Fos positive cells in the paraventricular nucleus of thalamus, posterior part (PVP) (**A**), and lateral/basolateral (LA/BL) complex of the amygdala (**C**). A statistically significant change in the number of c-Fos positive cells in the PVP (**B**) and LA/BL complex of the amygdala (**D**) in the controls, (Cont) and control rats treated with Olz (Cont + Olz), Clz (Cont + Clz) or Flx (Cont + Flx). Two independent investigators, blinded to the experimental conditions, counted the c-Fos positive cells. Significant differences between experimental groups, obtained from two-way ANOVA analyses, followed by Duncan’s post hoc test, are indicated as follows: ** *p* < 0.01, *** *p* < 0.001, compared to Cont rats. Scale bare 100 µm.

**Figure 4 pharmaceuticals-17-01527-f004:**
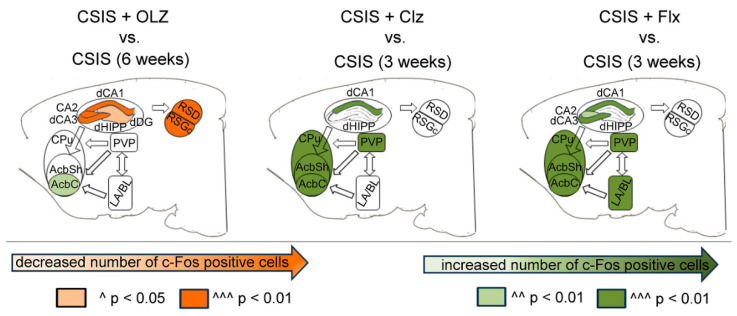
The schematic of rat brain sub/regions with decreased or increased number of c-Fos positive cells in the Olz-treated CSIS rats (lasting 3 weeks of 6-week CSIS), and Clz and Flx treatment (applied simultaneously with three weeks of CSIS), as compared to CSIS. Significant differences between experimental groups, obtained from two-way ANOVA analyses followed by Duncan’s post hoc test, are indicated as follows: ^ *p* < 0.05, ^^ *p* < 0.01, ^^^ *p* < 0.001, compared to CSIS rats. Used abbreviations are dHIPP—dorsal hippocampus; dCA1—dorsal cornu ammonis 1, CA2—cornu ammonis 2; dCA3—dorsal cornu ammonis 3; RSD—retrosplenial dysgranular cortex; RSGc—retrosplenial granular cortex, c region; PVP—paraventricular nucleus of thalamus, posterior part; LA/BL—lateral/basolateral complex of the amygdala; CPu—caudate putamen; AcbC—nucleus accumbens core; AcbSh—nucleus accumbens shell.

**Figure 5 pharmaceuticals-17-01527-f005:**
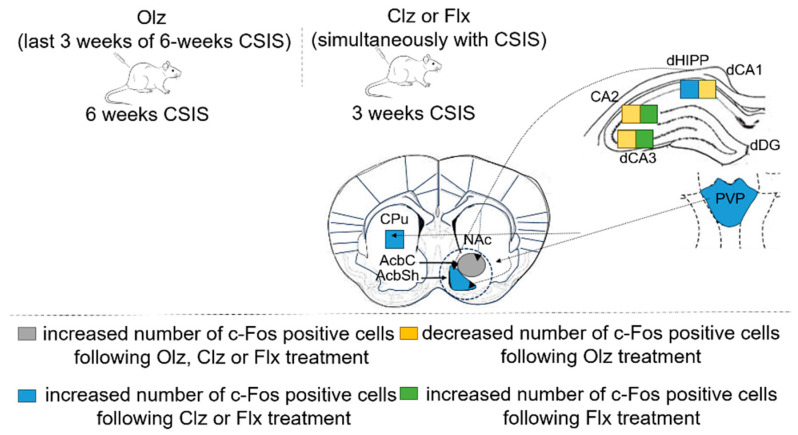
Schematic illustrating circuits of the rat brain sub/regions with overlapping patterns of c-Fos expression following effective Clz, Flx treatment (applied simultaneously with three weeks of CSIS), and Olz treatment in the CSIS rats (lasting 3 weeks of 6-week CSIS).

**Table 1 pharmaceuticals-17-01527-t001:** The overlapping rat brain sub/regions with matching pattern (↑ increased) of c-Fos expression following three and six weeks of chronic social isolation (CSIS) compared to controls: the restrosplenial cortex (RSC), restrosplenial granula cortex, c subregion (RSGc), restrosplenial dysgranular cortex (RSD), dorsal HIPP (dHIPP), dorsal dentate gyrus (dDG), paraventricular nucleus of the thalamus posterior part (PVP), lateral/basolateral (LA/BL) complex of the amygdala, caudate putamen (CPu), and nucleus accumbens shell (AcbSh). Significant differences between experimental groups (CSIS 3 weeks, CSIS 6 weeks), obtained from two-way ANOVA analyses followed by Duncan’s post hoc test are indicated as follows: * *p* < 0.05, ** *p* < 0.01, *** *p* < 0.001, compared to controls.

CSIS (Three and Six Weeks) vs. Controls				
Brain Region	Brain Subregion	↑c-Fos Expression		
		CSIS (3 Weeks) [60]	CSIS (3 Weeks) [58]	CSIS (6 Weeks) [15]
RSC	RSGc/RSD	** *p* < 0.01/ *** *p* < 0.001	*** *p* < 0.001	*** *p* < 0.001
dHIPP	dDG,	*** *p* < 0.001	* *p* < 0.05	*** *p* < 0.001
thalamus	PVP	** *p* < 0.01	*** *p* < 0.001	*** *p* < 0.001
amygdala	LA/BL complex of the amygdala	*** *p* < 0.001	*** *p* < 0.001	*** *p* < 0.001
dorsal striatum/NAc	CPu/AcbSh	* *p* < 0.05/ *** *p* < 0.001	* *p* < 0.05/ *** *p* < 0.001	*** *p* < 0.001

**Table 2 pharmaceuticals-17-01527-t002:** The overlapping brain sub/regions with matching pattern (↑ increased) of c-Fos expression in controls following treatment with antipsychotics olanzapine (Olz) and clozapine (Clz), and antidepressant fluoxetine (Flx).

Olz-, Clz- or Flx-Treated Controls vs. Controls			
Brain Region	Brain Subregion	Treatments	c-Fos Expression
thalamus	PVP	Olz, Clz or Flx	↑
amygdala	LA/BL complex of amygdala	Olz, Clz or Flx	↑
dorsal striatum	CPu	Olz or Clz	↑
NAc	AcbC	Olz or Clz	↑
NAc	AcbSh	Olz or Clz	↑
dHPP	dDG	Clz or Flx	↑
RSC	RSD	Clz or Flx	↑
RSC	RSGc	Clz or Flx	↑

**Table 3 pharmaceuticals-17-01527-t003:** Overlapping brain sub/regions with different pattern (↑ increased or ↓ decreased) of c-Fos expression in CSIS rats following treatment with antipsychotic olanzapine (Olz) (lasting 3 weeks of 6-week CSIS), and simultaneously with antipsychotic clozapine (Clz) and antidepressant fluoxetine (Flx).

Olz-, Clz- or Flx-Treated CSIS vs. CSIS			
Brain Region	Brain Subregion	Treatments	c-Fos Expression
NAc	AcbC	Olz, Clz or Flx	↑
dHIPP	dCA1	Clz or Flx/Olz	↑/↓
thalamus	PVP	Clz or Flx	↑
dorsal striatum/NAc	CPu/AcbSh	Clz or Flx	↑
dHIPP	CA2	Flx/Olz	↑/↓
dHIPP	dCA3	Flx/Olz	↑/↓

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
