# Peer review of "Mapping of c-Fos Expression in Rat Brain Sub/Regions Following Chronic Social Isolation: Effective Treatments of Olanzapine, Clozapine or Fluoxetine"

_pharmaceuticals, 2024, doi:10.3390/ph17111527_

Round 1
Reviewer 1 Report
Comments and Suggestions for Authors
The topic is interesting and the manuscript contains a lot of detailed information,
However, the conclusions are very general providing no insight into the complex topic reviewed in this manuscript. Therefore, the authors should rewrite conclusions.
Additionally, please provide citations in tables 1, 2 and 3 to help the readers to localize the data in original papers.
What is the source of images presented in Figure 3? Do the authors have a permission to use these images?
Author Response
Reviewer 1
The topic is interesting and the manuscript contains a lot of detailed information.
Response: We sincerely appreciate the time and effort you invested in reviewing our review. which have been fully addressed in the revised version of the review. These contributions have significantly helped enhance the quality of our work.
However, the conclusions are very general providing no insight into the complex topic reviewed in this manuscript. Therefore, the authors should rewrite conclusions.
Response: We thank you for the suggestion. We have revised the conclusions and provided more detail on the topics addressed in our revised review.
Additionally, please provide citations in tables 1, 2, and 3 to help the readers localize the data in the original papers.
Response: Thank you for this suggestion. We have included the appropriate reference number in Tables 1, 2, and 3 in the revised review. The following references are:
1.Stanisavljević, A.; Perić, I.; Bernardi, R.E.; Gass, P.; Filipović, D. Clozapine Increased C-Fos Protein Expression in Several Brain Subregions of Socially Isolated Rats. Brain Res. Bull. 2019, 152, 35–44, doi:10.1016/j.brainresbull.2019.07.005. Reference number 60.
2.Stanisavljević, A.; Perić, I.; Gass, P.; Inta, D.; Lang, U.E.; Borgwardt, S.; Filipović, D. Fluoxetine Modulates Neuronal Activity in Stress-Related Limbic Areas of Adult Rats Subjected to the Chronic Social Isolation. Brain Res. Bull. 2020, 163, 95–108, doi:10.1016/J.BRAINRESBULL.2020.07.021. Reference number 58.
3.Stanisavljević, A.; Perić, I.; Gass, P.; Inta, D.; Lang, U.E.; Borgwardt, S.; Filipović, D. Brain Sub/Region-Specific Effects of Olanzapine on c-Fos Expression of Chronically Socially Isolated Rats. Neuroscience 2019, 396, 46–65, doi:10.1016/j.neuroscience.2018.11.015. Reference number 15.
What is the source of the images presented in Figure 3? Do the authors have permission to use these images?
Response: Thank you for raising this question. We would like to clarify that the images shown in Figure 3 are our original photographs taken using a BTC light microscope equipped with a BIM 313T digital camera.
Reviewer 2 Report
Comments and Suggestions for Authors
Manuscript Review Pharmaceuticals-3286888-v1"Mapping of c-Fos expression in rat brain sub/regions following chronic social isolation: effective treatments of olanzapine, clozapine or fluoxetine" by A S Ilić and D Filipović.
In this review, the authors summarize the expression pattern of the immediate early gene c-Fos in various brain sub regions to identify neural circuits that underlie depressive- and anxiety-like behaviors in adult male rats following three and six weeks of chronic social isolation (CSIS) and in the presence of antipsychotics, olanzapine (Olz), and clozapine (Clz), and of fluoxetine (Flx) in CSIS relative to CSIS alone. They describe increased c-Fos immunoreactivity following three or six weeks of CSIS in the retrosplenial granular cortex, c-subregion, retrosplenial dysgranular cortex, dorsal dentate gyrus, paraventricular nucleus of the thalamus (posterior part), latero-basolateral complex of the amygdala, caudate putamen, and nucleus accumbens shell. They also summarize the increased activity of the nucleus accumbens core following exposure of CSIS rats either to Olz, Clz, and Flx treatments.
This quite exhaustive review is of interest for the readers. One could advise the authors to be more cautious in the pharmacological interpretation of the findings. Chronic isolation as well as chronic exposure to antipsychotics or to antidepressants may alter expression of various effectors. Therefore, this should be analyzed before concluding that such or such receptor may be involved.
In addition, when the authors are writing "Olanzapine (Olz) and clozapine (Clz) are atypical antipsychotics used to treat depressive and anxiety disorders [26,27], that act as antagonists at dopamine D2 and serotonin 5-HT2A/2C receptors [28]", they should see paper such as Shahid, et al. (Asenapine: a novel psychopharmacologic agent with a unique human receptor signature J Psychopharm 2009 vol. 23 pp. 65-73) showing that pharmacology of these compound is not as simple.
Similarly, when they say "Fluoxetine (Flx), a selective serotonin reuptake inhibitor, antagonizes the serotonin 5-HT2C receptor [34] preventing the reuptake of serotonin into presynaptic neurons which upregulates the amount of serotonin in the synaptic cleft [35]." They could read and discuss the paper by Sánchez, and Hyttel (Comparison of the effects of antidepressants and their metabolites on reuptake of biogenic amines and on receptor binding by Cell Mol Neurobiol 1999 vol. 19 pp. 467-89).
There are some typos e.g. p7 serotonine transporters line 17 from bottom. This should read serotonin transporter since a single gene exist for serotonin transporter.
Author Response
Reviewer 2
Manuscript Review Pharmaceuticals-3286888-v1"Mapping of c-Fos expression in rat brain sub/regions following chronic social isolation: effective treatments of olanzapine, clozapine or fluoxetine" by A S Ilić and D Filipović.
In this review, the authors summarize the expression pattern of the immediate early gene c-Fos in various brain sub regions to identify neural circuits that underlie depressive- and anxiety-like behaviors in adult male rats following three and six weeks of chronic social isolation (CSIS) and in the presence of antipsychotics, olanzapine (Olz), and clozapine (Clz), and of fluoxetine (Flx) in CSIS relative to CSIS alone. They describe increased c-Fos immunoreactivity following three or six weeks of CSIS in the retrosplenial granular cortex, c-subregion, retrosplenial dysgranular cortex, dorsal dentate gyrus, paraventricular nucleus of the thalamus (posterior part), latero-basolateral complex of the amygdala, caudate putamen, and nucleus accumbens shell. They also summarize the increased activity of the nucleus accumbens core following exposure of CSIS rats either to Olz, Clz, and Flx treatments.
Response: We sincerely appreciate the time and effort you dedicated to reviewing our review. We are grateful for the comments and valuable suggestions, which we have fully addressed in the revised review. These contributions have significantly enhanced the quality of our work.
This quite exhaustive review is of interest for the readers. One could advise the authors to be more cautious in the pharmacological interpretation of the findings. Chronic isolation as well as chronic exposure to antipsychotics or to antidepressants may alter expression of various effectors. Therefore, this should be analyzed before concluding that such or such receptor may be involved.
In addition, when the authors are writing "Olanzapine (Olz) and clozapine (Clz) are atypical antipsychotics used to treat depressive and anxiety disorders [26,27], that act as antagonists at dopamine D2 and serotonin 5-HT2A/2C receptors [28]", they should see paper such as Shahid, et al. (Asenapine: a novel psychopharmacologic agent with a unique human receptor signature J Psychopharm 2009 vol. 23 pp. 65-73) showing that pharmacology of these compound is not as simple.
Response: We thank you for valuable suggestion regarding the pharmacology aspects of olanzapine and clozapine. Based on your suggestion, we have added the suggested reference to the Introduction section and highlighted the complexity of the pharmacological profiles of these atypical antipsychotics.
Similarly, when they say "Fluoxetine (Flx), a selective serotonin reuptake inhibitor, antagonizes the serotonin 5-HT2C receptor [34] preventing the reuptake of serotonin into presynaptic neurons, which upregulates the amount of serotonin in the synaptic cleft [35]." They could read and discuss the paper by Sánchez, and Hyttel (Comparison of the effects of antidepressants and their metabolites on reuptake of biogenic amines and on receptor binding by Cell Mol Neurobiol 1999 vol. 19 pp. 467-89).
Response: Thank you for your valuable suggestion. We agree that adding this reference would strengthen the pharmacological discussion and provide further insight into the mechanisms of fluoxetine. Hence, we have added reference Sánchez and Hyttel (1999) in revised review, which provides valuable comparative analysis on the receptor binding profiles and reuptake inhibition mechanisms of fluoxetine.
There are some typos e.g. p7 serotonine transporters line 17 from bottom. This should read serotonin transporter since a single gene exist for serotonin transporter
Response: We thank you for pointing out this typographical error. In the revised review, we have corrected "serotonin transporters" to "5-HTT" to reflect the correct terminology, as a single gene exists for the serotonin transporter.
Reviewer 3 Report
Comments and Suggestions for Authors The manuscript focuses on the expression of c-Fos protein as a marker for brain activity in rat models that have been subjected to CSIS, with treatment with olanzapine, clozapine, and fluoxetine. The present study is significant enough to identify brain subregions that are associated with depressive and anxiety-like behaviors in rats, with the intention of informing neuroanatomical targets for the potential treatment of these disorders. Some comments are listed below for consideration by the authors: 1. The introduction sets the stage for the purpose of the study, providing background information on c-Fos as a neuronal marker and putting it into context in relation to the study of stress responses and treatments. However, more information related to the recent literature concerning the involvement of chronic social stress models for translational research into depression and anxiety treatments could focus more on the manuscript. Of course, references to recent work on antipsychotic and antidepressant efficacy regarding neurocircuitry would progress the depth of the introduction. 2. The methodology is comprehensive but does not account for the reproducibility of data collection and analysis. For example, quantification parameters utilized for c-Fos immunoreactivity need to be provided, such as image preparation, which is lacking and should be so for transparency. Ethical guidelines regarding animal use are not discussed and should be accounted for here to confirm adherence to standards on research. 3. The results section provides a good analysis of the brain subregions activated by CSIS and pharmacological treatments. However, some of the figures, especially the schematic figures, need to be redesigned for clarity. These results would be further strengthened by the inclusion of additional statistical analyses confirming the observed trends in c-Fos expression, particularly for the comparative effects of olanzapine, clozapine, and fluoxetine. 4. This discussion sets the findings in the context of the current literature on c-Fos and neuroanatomical responses to stress. However, some statements were not fully supported by data. This section would be even stronger if the limitations-the specificity of c-Fos as a cellular marker for activation and possible confounding factors-were subject to more critical scrutiny. Further improvement in the analytical level of the manuscript would ensue from discussing alternative explanations of the pharmacological data. 5. Statistical comparisons of pharmacological effects were included to strengthen the arguments. 6. Major Issues: • The manuscript needs to clarify the form of adherence to ethical guidelines, further mentioning any information on reproducibility, mainly regarding the analysis of images or data collection. • There is underreporting of statistical analyses that may impact or question the robustness of the conclusions. 7. MINOR ISSUES: • Some terms are technical and wording modifications can improve readability. • Figures need enhanced quality and clarity to match the reported findings.Author Response
Reviewer 3
The manuscript focuses on the expression of c-Fos protein as a marker for brain activity in rat models that have been subjected to CSIS, with treatment with olanzapine, clozapine, and fluoxetine. The present study is significant enough to identify brain subregions that are associated with depressive and anxiety-like behaviors in rats, with the intention of informing neuroanatomical targets for the potential treatment of these disorders.
Response: We truly appreciate the time and effort you put into reviewing our review. Your comments and valuable suggestions have been fully addressed in this revised version, and they have greatly helped us improve the quality of our review.
Some comments are listed below for consideration by the authors:
- The introduction sets the stage for the purpose of the study, providing background information on c-Fos as a neuronal marker and putting it into context in relation to the study of stress responses and treatments. However, more information related to the recent literature concerning the involvement of chronic social stress models for translational research into depression and anxiety treatments could focus more on the manuscript. Of course, references to recent work on antipsychotic and antidepressant efficacy regarding neurocircuitry would progress the depth of the introduction.
Response: Thank you for your helpful feedback. As you requested, we have expanded the Introduction section with more recent literature (references 40–57) on chronic social stress models in translational research for the treatment of depression and anxiety. Additionally, we have added references to recent studies on the efficacy of antipsychotics and antidepressants in relation to neurocircuitry.
- The methodology is comprehensive but does not account for the reproducibility of data collection and analysis. For example, quantification parameters utilized for c-Fos immunoreactivity need to be provided, such as image preparation, which is lacking and should be so for transparency. Ethical guidelines regarding animal use are not discussed and should be accounted for here to confirm adherence to standards on research.
Response: We thank you for the comments. We agree that additional detail on the quantification parameters for c-Fos immunoreactivity would enhance transparency. In the revised review we have included a description of image preparation and the quantification of c-Fos positive cells.
In the revised review in the section Institutional Review Board Statement, we have included the ethical license ensuring compliance with the highest standards for animal welfare in research. This review is based on findings from our three previously published papers:
1) Stanisavljević, A.; Perić, I.; Gass, P.; Inta, D.; Lang, U.E.; Borgwardt, S.; Filipović, D. Brain Sub/Region-Specific Effects of Olanzapine on c-Fos Expression of Chronically Socially Isolated Rats. Neuroscience 2019, 396, 46–65, doi:10.1016/j.neuroscience.2018.11.015. Reference number 15.
2) Stanisavljević, A.; Perić, I.; Gass, P.; Inta, D.; Lang, U.E.; Borgwardt, S.; Filipović, D. Fluoxetine Modulates Neuronal Activity in Stress-Related Limbic Areas of Adult Rats Subjected to the Chronic Social Isolation. Brain Research Bulletin 2020, 163, 95–108, doi:10.1016/j.brainresbull.2020.07.021. Reference number 58.
3) Stanisavljević, A.; Perić, I.; Bernardi, R.E.; Gass, P.; Filipović, D. Clozapine Increased C-Fos Protein Expression in Several Brain Subregions of Socially Isolated Rats. Brain Research Bulletin 2019, 152, 35–44, doi:10.1016/j.brainresbull.2019.07.005. Reference number 60.
Therefore, we have cited the ethical guidelines regarding animal use from these prior manuscripts, which certify adherence to animal welfare standards throughout our research.
- The results section provides a good analysis of the brain subregions activated by CSIS and pharmacological treatments. However, some of the figures, especially the schematic figures, need to be redesigned for clarity. These results would be further strengthened by the inclusion of additional statistical analyses confirming the observed trends in c-Fos expression, particularly for the comparative effects of olanzapine, clozapine, and fluoxetine.
Response: Thank you for this suggestion. We would like to emphasize that the presented results of the effects of three-week and six-week CSIS, as well as the effects of olanzapine, clozapine, and fluoxetine in both control and CSIS rats in our review, represent findings from three separate previously published studies from our laboratory-References numbers 15, 58 and 60. (please see previous response).
According to your suggestion, we have added statistical analysis obtained from two-way ANOVA followed by Duncan's post hoc test in Table 1, which shows the overlapping rat brain subregions with matching patterns of c-Fos expression after three and six weeks of CSIS, compared to the controls. Additionally, we have included statistical analyses from the same tests in Figure 3, which compares the effects of olanzapine, clozapine, and fluoxetine in controls, separately against the control group. We have also revised Figure 4 to enhance its clarity and included the relevant statistical analyses. For Figure 5, no additional statistical analyses were included, as this figure is intended solely to illustrate the neural circuits between the overlapping brain regions already presented in Figure 4.
- This discussion sets the findings in the context of the current literature on c-Fos and neuroanatomical responses to stress. However, some statements were not fully supported by data. This section would be even stronger if the limitations-the specificity of c-Fos as a cellular marker for activation and possible confounding factors-were subject to more critical scrutiny. Further improvement in the analytical level of the manuscript would ensue from discussing alternative explanations of the pharmacological data.
Response: Thank you for your comments. In the revised review, the limitations of using c-Fos, as a cell activation marker, have added in Introduction section. Additionally, we have added alternative explanations for the pharmacological data in Discussion section.
- Statistical comparisons of pharmacological effects were included to strengthen the arguments.
Response: We thank you for suggestion. We have included statistical comparisons obtained from a two-way ANOVA followed by Duncan's post hoc test in both Figure 3 and Figure 4 to further clarify the pharmacological effects of clozapine, olanzapine, and fluoxetine treatments in controls (Figure 3) and CSIS rats (Figure 4).
- Major Issues: • The manuscript needs to clarify the form of adherence to ethical guidelines, further mentioning any information on reproducibility, mainly regarding the analysis of images or data collection. • There is underreporting of statistical analyses that may impact or question the robustness of the conclusions.
Response: We thank you for your comments. We have added the Ethical guidelines in the section Institutional Review Board Statement, as well as information regarding analysis of images or data, and statistical analyses in the revised review.
- MINOR ISSUES: • Some terms are technical and wording modifications can improve readability. • Figures need enhanced quality and clarity to match the reported findings.
Response: We thank you for suggestions. We have revised the text for better readability and comprehension. Also, we have enhanced the resolution and added statistical information in Figures to ensure they represent the findings as reported.
Round 2
Reviewer 3 Report
Comments and Suggestions for Authors
The work is highly enhanced